# Light and electron microscopic features of preclinical pseudoexfoliation syndrome

**Yanin Suwan**[1], **Tuangporn Kulnirandorn**[1], **Ursula Schlötzer-Schrehardt**[2], **Sattawat Wongchaya**[1], **Purit Petpiroon**[1], **Wasu Supakontanasan**[1]*, **Apichat Tantraworasin**[3,4], **Chaiwat Teekhasanee**[1]

1 Glaucoma Service, Department of Ophthalmology, Ramathibodi Hospital, Mahidol University, Bangkok, Thailand, 2 University Clinic Erlangen, Friedrich-Alexander-Universität Erlangen-Nürnberg, Erlangen, Germany, 3 Department of Surgery and Clinical Epidemiology and Clinical Statistic Center, Faculty of Medicine, Chiang Mai University, Chiang Mai, Thailand, 4 Clinical Surgical Research Center, Chiang Mai University, Chiang Mai, Thailand

\* yanin.suwan@gmail.com

**Data Availability Statement:** All data files are publicly available from the figshare repository

## Abstract

### Purpose

This study sought to explore the features of the anterior lens capsule in patients with preclinical pseudoexfoliation syndrome (pPEX) via light microscopy (LM) and transmission electron microscopy (TEM).

### Design

Cross-sectional, prospective, and observational case series.

### Methods

We recruited consecutive patients with and without pPEX who underwent routine cataract surgery at Ramathibodi Hospital, between April 2018 and November 2020. pPEX can be characterized by pigmented spoke-wheel deposition (P) on the anterior lens capsule, mid-periphery cleft/lacunae (C), faint central disc present within the photopic pupil (D), white-spoke pattern (W) noted at the midperiphery, and a combination of at least two signs (Co). LM and TEM were used to examine anterior lens capsule specimens for the presence of characteristic pseudoexfoliation material (PXM). The features of the anterior lens capsule in pPEX explored via LM and TEM were recorded.

### Results

This study included a total of 96 patients (101 excised anterior lens capsules); among them, 34 (35 excised anterior lens capsules) exhibited pPEX signs (pPEX group) but 62 (66 excised anterior lens capsules) did not (control group). The patients had a mean age of 74 ± 7 (range, 58–89) years. LM and TEM revealed no definite PXM in any patient. In the pPEX group, LM analysis identified two capsule specimens with suspected PXM; PXM precursors were observed in 1 of the 34 excised capsule specimens analyzed via TEM. Furthermore, 39 eyes (59.09%) exhibited signs of true exfoliation syndrome (TEX) in LM analysis

(https://doi.org/10.6084/m9.figshare.22179802.
v1).

**Funding:** Ramathibodi Foundation (RF_61056) The
funders had no role in study design, data collection
and analysis, decision to publish, or preparation of
the manuscript.

**Competing interests:** The authors have declared
that no competing interests exist.

(12.82%, 25.64%, 10.26%, 10.26%, and 41.03% for patients exhibiting P, D, C, W, and Co, respectively). However, no TEX signs were observed in the control group. We found that the anterior lens capsules exhibiting C and D were significantly associated with TEX (odds ratio = 5.4 and 7.9; P = 0.007 and 0.004, respectively).

## Conclusions

LM analysis revealed no definite PXMs were detected in the excised anterior lens capsules, whereas TEM analysis showed PXM precursors in one specimen (2.94%). Notably, a significant association was observed between C and D signs and TEX.

## Introduction

Pseudoexfoliation syndrome (PEX) is characterized by the production and progressive accumulation of a fibrillar extracellular material in various systemic and ocular structures. PEX is the most common identifiable cause of open-angle glaucoma [1]. Previous studies have reported the detection of pseudoexfoliation material (PXM) in extraocular structures such as the extraocular muscles, lung, heart, liver, gallbladder, skin, kidney, and meninges [2, 3]. PEX is an age-related disease [4], with patients often clinically presenting with unilateral eye involvement. Considering uninvolved eye has an 81% likelihood of being affected ultrastructurally, evidence suggests that progression to bilateral eye involvement occurs in up to 50% of the patients within 5–10 years after diagnosis, particularly in older patients [5]. PEX is the commonest cause of secondary glaucoma world-wide and is also a prognostic factor for progression of open-angle glaucoma. Patients with PEX have an approximately 40% risk of the onset or development of ocular hypertension or glaucoma within 10 years [1, 6]. Glaucoma in PEX tends to have more severe clinical course, showing a prognosis worse than that of primary open-angle glaucoma [1, 7]. In addition, the disease causes lens subluxation; angle-closure glaucoma; blood–aqueous barrier impairment; and cataract surgery-related complications such as zonular dialysis, capsular rupture, and vitreous loss [8].

PEX is diagnosed by evaluating the anterior segment structures using slit-lamp biomicroscopy. Several studies have reported that patients with no evidence of PXM in either eye but exhibiting signs related to the loss and dispersion of pigment from the iris pigment epithelium can be diagnosed with or suspected of having preclinical PEX (pPEX) [9]. Other signs of pPEX include pigment loss from the pupillary ruff, pigment deposition on the iris sphincter region, peripupillary transillumination defects, moderate to dense pigment deposition in the trabecular meshwork, incomplete mydriasis, and pigment shower after mydriasis.

Early recognition of PEX is crucial for identifying patients at a risk of rapidly progressing glaucoma development and possible surgical complications. Some signs of pPEX have been studied in the past. According to Prince et al., eyes with signs associated with pigment dispersion, such as pupillary ruff defects; iris sphincter transillumination defects; and pigment deposition on the iris sphincter, iris surface, and trabecular meshwork, may indicate early signs of PEX in the absence of PXM [9]. To date, studies on the ultrastructural alteration of the anterior capsule in eyes exhibiting the signs of pPEX are limited. Therefore, we aimed to identify the correlation between pPEX signs and light microscopy (LM) and transmission electron microscopy (TEM) findings.

## Methods

This cross-sectional, observational study was approved by the Ramathibodi Institutional Review Board/Ethics Committee (MURA 2018/99). This study was conducted in accordance with the Health Insurance Portability and Accountability Act as well as the tenets of the Declaration of Helsinki. Written informed consent was obtained from all participants.

## Participants

This study enrolled consecutive patients aged >50 years who visited the ophthalmology department of Ramathibodi Hospital, Mahidol University, Thailand, with a complaint of symptomatic cataract with and without the signs of pPEX between April 2018 and December 2018.

pPEX was defined as the presence of the following signs without clinically identifiable PXM on the anterior lens capsule or pupillary margin in either eye: (1) pigmented spoke-wheel deposition (P) on the anterior lens capsule (Fig 1A), (2) faint central disc (D) present within the photopic pupil (Fig 1B), (3) midperiphery cleft/lacunae (C; Fig 1C), and (4) white-spoke pattern (W) noted at the midperiphery of the lens capsule (Fig 1D). We hypothesized that P, D, and C originated from continuous rubbing of PXM on the posterior iris against anterior lens, whereas W represents early stage of PXM formation.

We included patients who underwent complete ocular examination. This study excluded patients with a history of eye trauma, uveitis, pigmentary dispersion, media opacities, conditions that affected the anterior chamber and angle examination, and laser or surgical

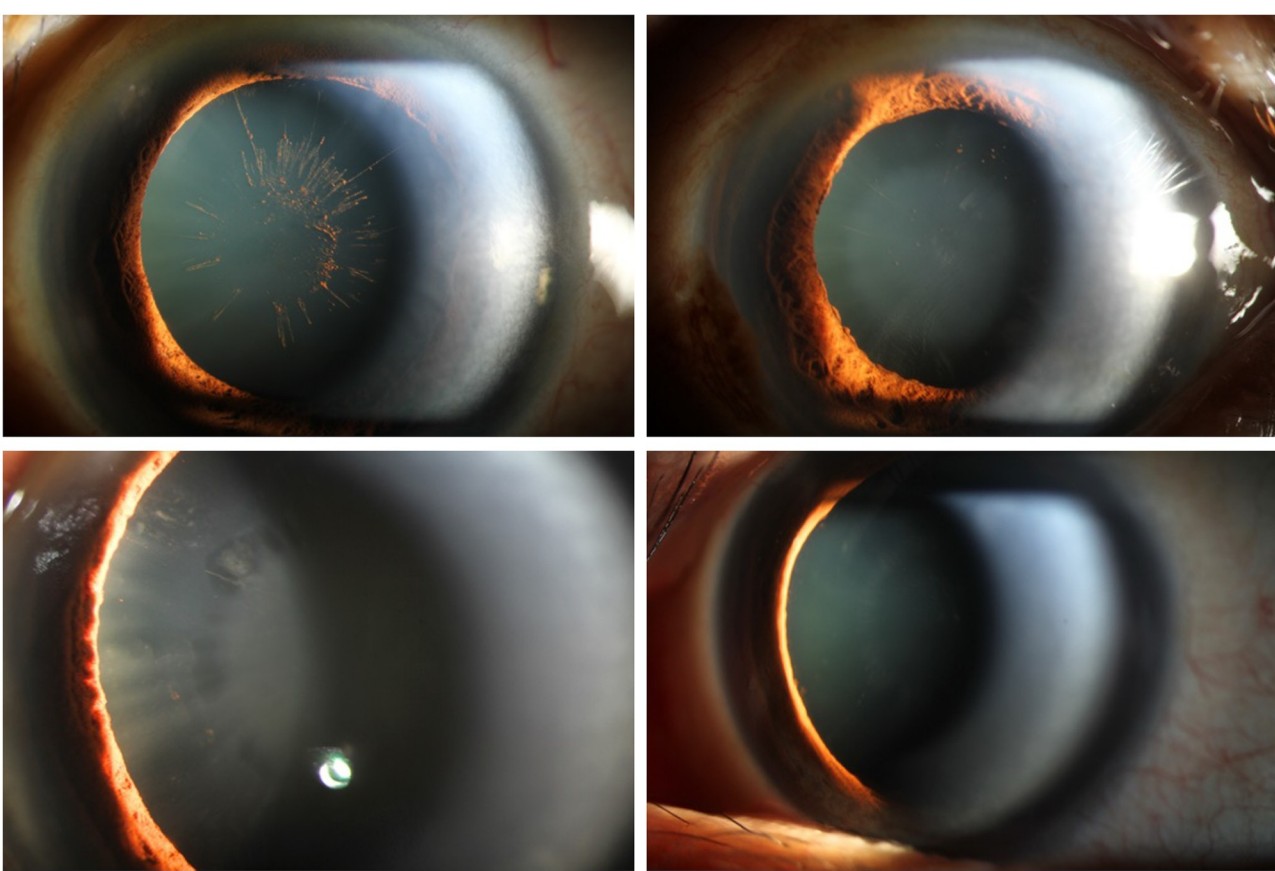

**Fig 1. Anterior segment photographs of preclinical pseudoexfoliation syndrome.** (A) Pigmented spoke wheel deposition on anterior lens capsule (B) Faint central disc within photopic pupil (C) Midperiphery cleft/lacunae (D) White-spoke pattern at midperiphery.

treatment; those with the presence of PXM on the pupillary margin or anterior lens capsule; and those who were unwilling to participate.

The enrolled patients were examined using slit-lamp biomicroscopy as part of a detailed preoperative examination. The anterior capsular surface was photographed using a photo slit lamp (Haag-Streit BX900, Haag-Streit AG, Switzerland). Clinical data, including age, sex, associated eye disease, and true exfoliation syndrome (TEX) stage (if present), were collected.

## Surgical procedures

The enrolled patients underwent phacoemulsification and intraocular lens implantation. Capsulorhexis was performed using a 27-G needle cystotome and capsulorhexis forceps after trypan blue staining of the anterior lens capsule. Extensive care was taken to avoid iatrogenic injuries. After the completion of capsulorhexis, we collected the anterior lens capsule specimens by flushing with balanced salt solution combined with the pushing of the posterior wound lip through the temporal clear corneal incision. The delivered capsule was then spread into the original shape and position and fixed with glutaraldehyde. All specimens were immediately cut into halves and fixed in 1% glutaraldehyde at 4˚C and processed for LM or TEM analyses or both. No complications were noted during the surgery and follow-up.

## Specimen preparation

**LM.**   Each half was folded and stacked into three to four layers. Capsular tissue was embedded in paraffin wax and then cut radially. A total of 14 ribbons were taken out via serial 3-μm-thin microtome sectioning and allowed to float on the surface of a water bath. Seven randomly collected ribbons were each placed on glass slides. We prepared five glass slides per eye while sectioning proceeded into the deeper layers of the block. Thereafter, the slides were stained with hematoxylin and eosin and studied using LM (Olympus BX61 with DP72 camera and Olympus cellSens Dimension V1.18 software, Tokyo, Japan). The processed anterior lens capsule specimens were examined using LM at 4×, 10×, 40×, and 100× magnifications. The presence of PXMs and other findings were recorded.

**TEM.**   Since some capsules had been lost and fallen dry during transportation, a total of 34 anterior lens capsules were sent for TEM analysis. The specimens were fixed in 2.5% glutaraldehyde in 0.1 M phosphate buffer, postfixed in 2% buffered osmium tetroxide, dehydrated in graded alcohol concentrations, and embedded in epoxy resin following standard protocols. Moreover, 1-μm-thin tissue sections were stained with toluidine blue for orientation. Ultrathin sections were stained with uranyl acetate and lead citrate and examined using TEM (EM 906E; Carl Zeiss Microscopy, Oberkochen, Germany).

## Statistical analysis

Descriptive statistics were presented as numbers and percentages for normally distributed variables. Mixed-effects multilevel regression analysis stratified according to laterality was performed to determine the association between pPEX signs and TEX. All statistical analyses were performed using SPSS (version 22.0; IBM Corp., Armonk, NY, USA). P values of <0.05 were considered significant.

## Results

This study evaluated a total of 107 anterior lens capsules from 102 patients. Six specimens were excluded (one because of the poor quality of the tissue section and the other five because of a history of laser peripheral iridotomy performed before the diagnosis of pPEX).

**Table 1. Clinical, light, and electron microscopic features of anterior lens capsules in preclinical pseudoexfoliation syndrome.**

| Preclinical PEX Signs | LM findings | | | TEM findings | | |
|---|---|---|---|---|---|---|
| | N = 66, n (%) | TEX (N = 39, 59.09%), n (%) | PEX | Total specimen sent (N = 34), n (%) | TEX | PEX (N = 1), n (%) |
| P | 19 (28.79) | 5 (26.31) | 0 | 13 (38.23) | 0 | 0 |
| D | 13 (19.70) | 10 (76.92) | 0 | 6 (17.65) | 0 | 0 |
| C | 7 (10.61) | 4 (57.14) | 0 | 1 (2.94) | 0 | 1* (100) |
| W | 4 (6.06) | 4 (100) | 0 | 2 (5.88) | 0 | 0 |
| Combination of at least two signs | 23 (34.85) | 16 (69.57) | 0 | 12 (35.29) | 0 | 0 |
| • P+D | 10 (43.48) | 5 (50) | | 5 (41.67) | | |
| • P+C | 4 (17.40) | 4 (100) | | 2 (16.67) | | |
| • P+W | 1 (4.35) | 1 (100) | | 1 (8.33) | | |
| • D+C | 5 (21.74) | 4 (80) | | 4 (33.33) | | |
| • D+W | 1 (4.35) | 1 (100) | | 0 | | |
| • P+D+C | 2 (8.70) | 1 (50) | | 0 | | |

*TEM study showed PXM precursor.

N, number of eyes; C, midperiphery cleft/lacunae; Co, combination of at least two signs of preclinical pseudoexfoliation syndrome; D, faint central disc present within the photopic pupil; LM, light microscopy; P, pigmented spoke-wheel deposition on the anterior lens capsule; PEX, pseudoexfoliation syndrome; TEM, transmission electron microscopy; TEX, true exfoliation syndrome; W, white-spoke pattern noted at the midperiphery of the lens capsule.

Finally, a total of 101 anterior lens capsule specimens were obtained from a total of 96 patients (45 men and 56 women). The capsules were divided into two groups: 66 capsules in the pPEX group (capsules that exhibited pPEX signs) and 35 capsules in the control group (capsules that did not exhibit pPEX signs). The mean age was 74 ± 7 (range 58–89) years. Eight patients had coexisting clinical TEX (slit-lamp examination showing capsular delamination). The clinical, LM, and TEM features of pPEX are shown in Table 1.

Among the signs explored, P was the most common in this study, followed by D, C, and W. In addition, P combined with D was mostly found in the patients exhibiting the combination of at least two signs.

## LM analysis results

LM analysis revealed no definite PXM, although suspected PXM was observed in two of the excised capsule specimens in patients exhibiting P. The suspected materials were noted as a group of nodule-like eosinophilic lesions bulging from the anterior surface of the capsule (Fig 2). However, the findings were not exactly the same as those of classic PXM, which show a bush-like pattern of eosinophilic PXM on the surface of the anterior lens (Fig 3) [13]. More than half of specimens in the pPEX group (39 of 66; 59.09%) exhibited the signs of TEX (e.g., capsular delamination, Fig 4A; capsular lamination, Fig 4B; superficial cyst, Fig 4C; long tapering edge, Fig 4D; and vesicular change in the capsule layers, Fig 4E). No microscopic findings of TEX were noted in the control group. Table 2 presents the LM findings of the specimens obtained from patients exhibiting TEX signs.

A significant association was observed between pPEX signs and microscopic findings of TEX. Patients exhibiting D and C signs showed a 3.0- and 4.7-fold increase in the risk of TEX development, which was statistically significant ($p = 0.046$ and $0.020$, respectively). Patients exhibiting the P sign showed a 1.89-fold increase in the risk of TEX development, which was not statistically significant ($p = 0.232$). However, TEX was detected in all patients exhibiting W. Table 3 shows the association between the signs of clinical pPEX and TEX.

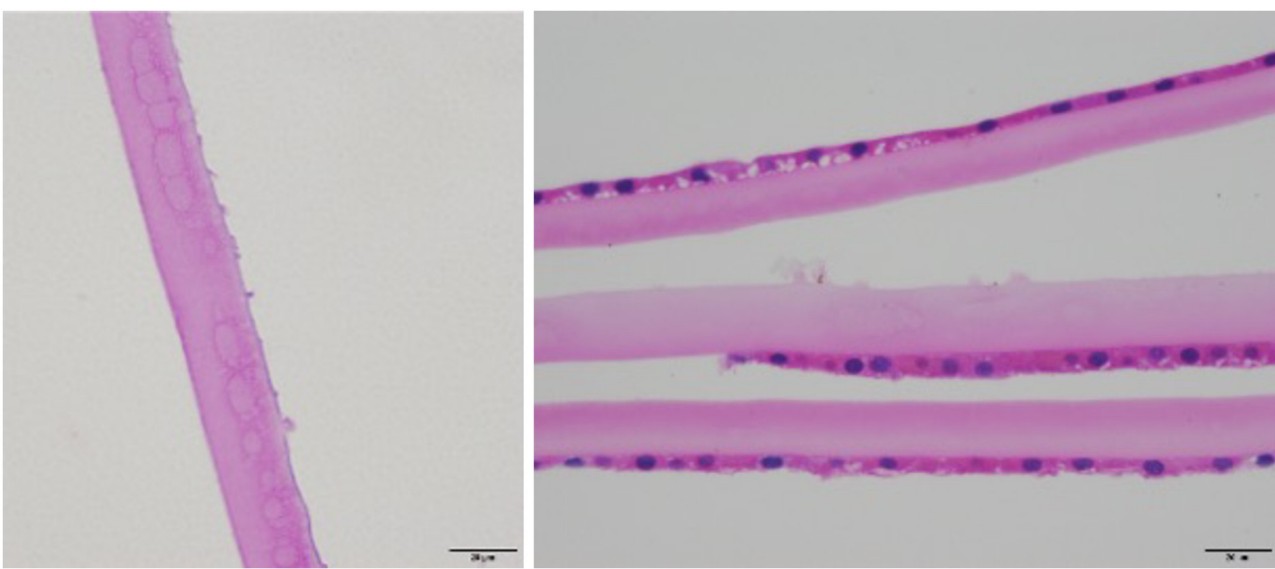

**Fig 2. Light microscopic photographs.** Suspected PXM on the anterior lens capsule. (A) The excised capsule with H&E staining showed eosinophilic stained materials bulging from the anterior lens capsule which suspected to be PXM. (B) This specimen also showed eosinophilic materials bulging from anterior lens capsule surface in one strip, while the other two strips showed smooth surface. There was some non-specific cystic change in lens capsule epithelium.

## TEM analysis results

A total of 34 capsule specimens were examined via TEM at two to three levels of sectioning. Majority of the specimens (n = 12) revealed a roughened and loosely structured capsular surface, partly with granular inclusions, which was prone to desquamation from the capsule in

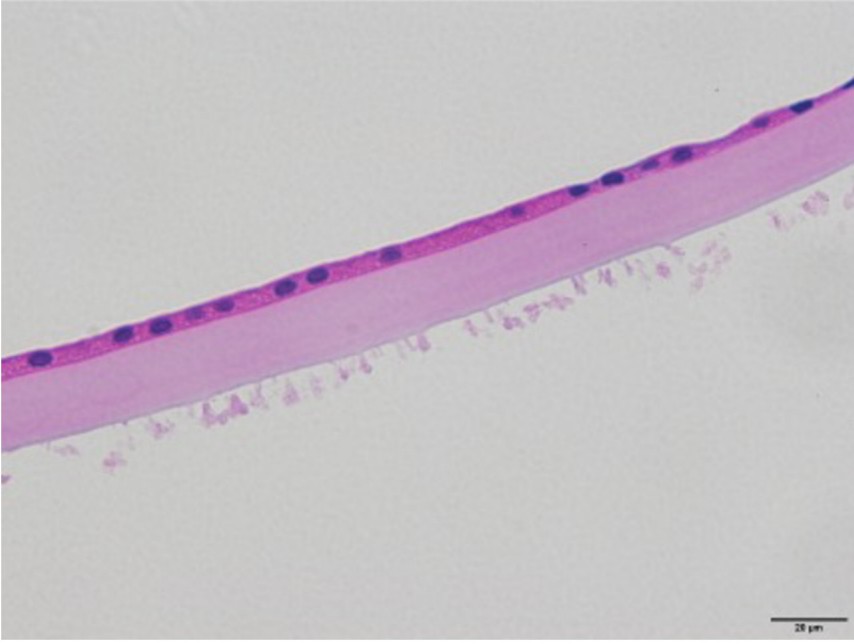

**Fig 3. Light microscopic photograph demonstrates classic bush-like deposition of pseudoexfoliation material.**

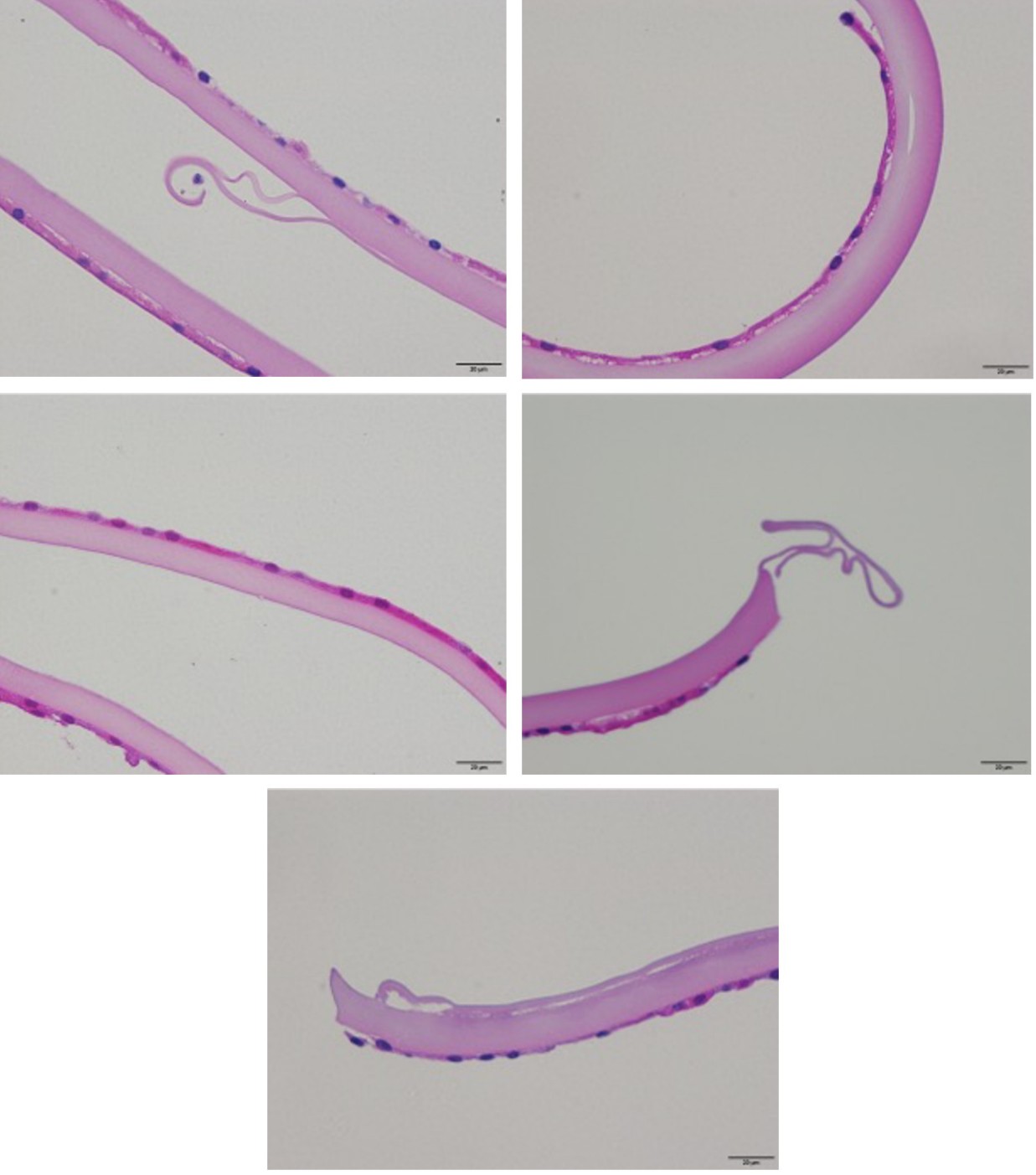

**Fig 4. Light microscopic photographs of true exfoliation syndrome.** (A) The specimen shows different thickness along the capsule with partial thickness capsular double delamination. (B) The excised capsule shows a space (black arrow) and multiple thin lines (red arrow) within the capsular propia which called capsular lamination. The lens epithelium showed vesicular change. (C) Superficial cyst within anterior lens capsule (D) Tapering edge of the delaminated membrane peripheral to the capsulorhexis with beveled down cut end of capsular propia (E) Vesicular change in the capsule. Coalescence of theses vesicles may progress to capsular delamination.

**Table 2. Light microscopic findings in true exfoliation syndrome group.**

| Light microscopic findings of TEX | Number (%) |
|---|---|
| Capsular delamination | 22 (56.41) |
| Capsular lamination | 7 (17.95) |
| Superficial cyst | 15 (38.46) |
| Long tapering edge | 17 (43.59) |
| Vesicular change within the capsule | 19 (48.72) |
| Pigment | 14 (35.9) |

TEX, true exfoliation syndrome.

patches (Fig 5A and 5B). In eight other specimens, zonular fiber remnants forming the zonular lamella were found to be attached to and have merged into this roughened surface layer in the midperipheral regions. Pigment granules were frequently noted to be adhered to the zonular fibrils (Fig 5C and 5D). Three specimens showed the deposition of single microfibrils, which might have been derived from the zonular fibers, on the roughened capsule surface (Fig 5E). Remnants of the iris pigment epithelium on the capsule surface, indicating the formation of posterior synechiae, were detected in one specimen (Fig 5F). Layer formation within the superficial capsule with a vacuolar interface between layers was evident in two specimens; focal splitting and desquamation of the most superficial layer along the interface in these cases was consistent with the clinical signs of TEX (Fig 6A and 6B). Finally, a smooth and compact capsular surface without any evident abnormalities was found in 8 of the 34 specimens (Fig 6C). Precursor PXM, which comprises amorphous and microfibrillar components and forms nodular deposits on the capsule surface, was observed in another specimen (Fig 6D). This capsule was clinically classified into the C group. LM analysis of this specimen revealed no PXM. The suspected PXM found in two of the excised capsule specimens in LM analysis were not detected in TEM analysis.

## Discussion

In the present study we found no association between pPEX signs and microscopic findings of PXM in the anterior lens capsule. Only 2 out of 66 capsules were suspected to have PXM, which could not be definitely differentiated from the zonular stumps in LM analysis. In TEM analysis, 1 of the 34 excised capsules exhibited PXM precursors.

In a previous study, Mardin et al. [10] examined the anterior lens capsule specimens of patients who underwent extracapsular cataract extraction using TEM; they detected PXM precursors in 20 of 35 (57.1%) patients who were clinically suspected to have PEX. Their results showed a stronger correlation between pPEX signs and the detection of PXM in TEM analysis

**Table 3. Association between the signs of preclinical pseudoexfoliation and true exfoliation syndromes.**

| Clinical characteristics | Odds ratio | 95% CI | P value |
|---|---|---|---|
| P | 1.897 | 0.664, 5.418 | 0.232 |
| D | 3.049 | 1.019, 9.121 | 0.046 |
| C | 4.731 | 1.279, 17.505 | 0.020 |
| Age | 1.116 | 1.031, 1.207 | 0.006 |

P, pigmented spoke-wheel deposition; D, faint central disc present within the photopic pupil; C, midperiphery cleft/lacunae, 95% CI, 95% confidence interval.

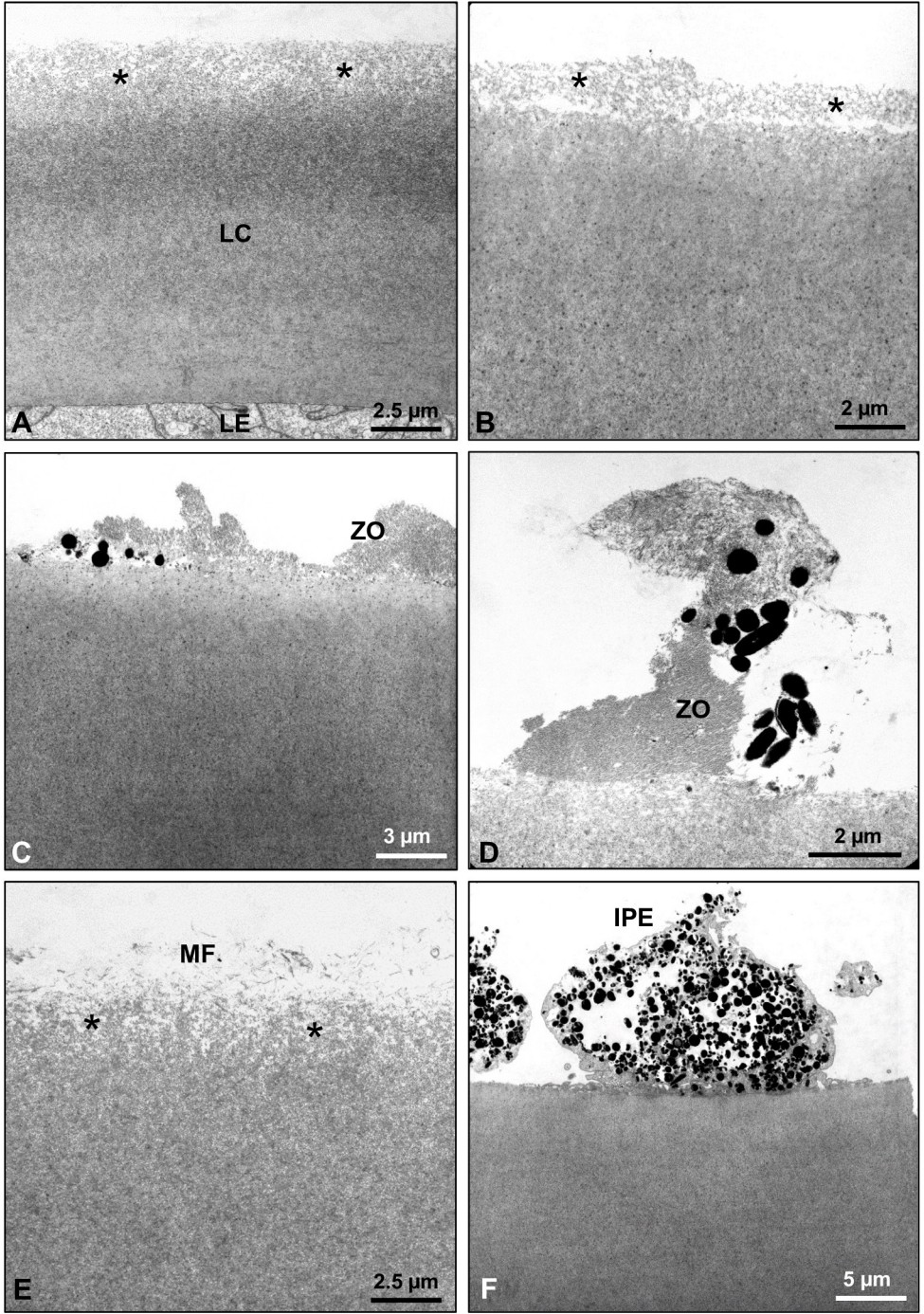

**Fig 5. Transmission electron micrographs of lens capsule specimens showing various alterations of the capsular surface.** (A, B) Roughened and loosely structured surface layer (asterisks) of the lens capsule (LC), which was focally detaching from the capsule. (C, D) Remnants of zonular fibers (ZO) forming the zonular lamella with adherent pigment granules. (E) Deposition of microfibrils (MF) onto the roughened capsule surface (asterisks). (F) Remnants of iris pigment epithelium (IPE) on the capsule surface. (LE, lens epithelium).

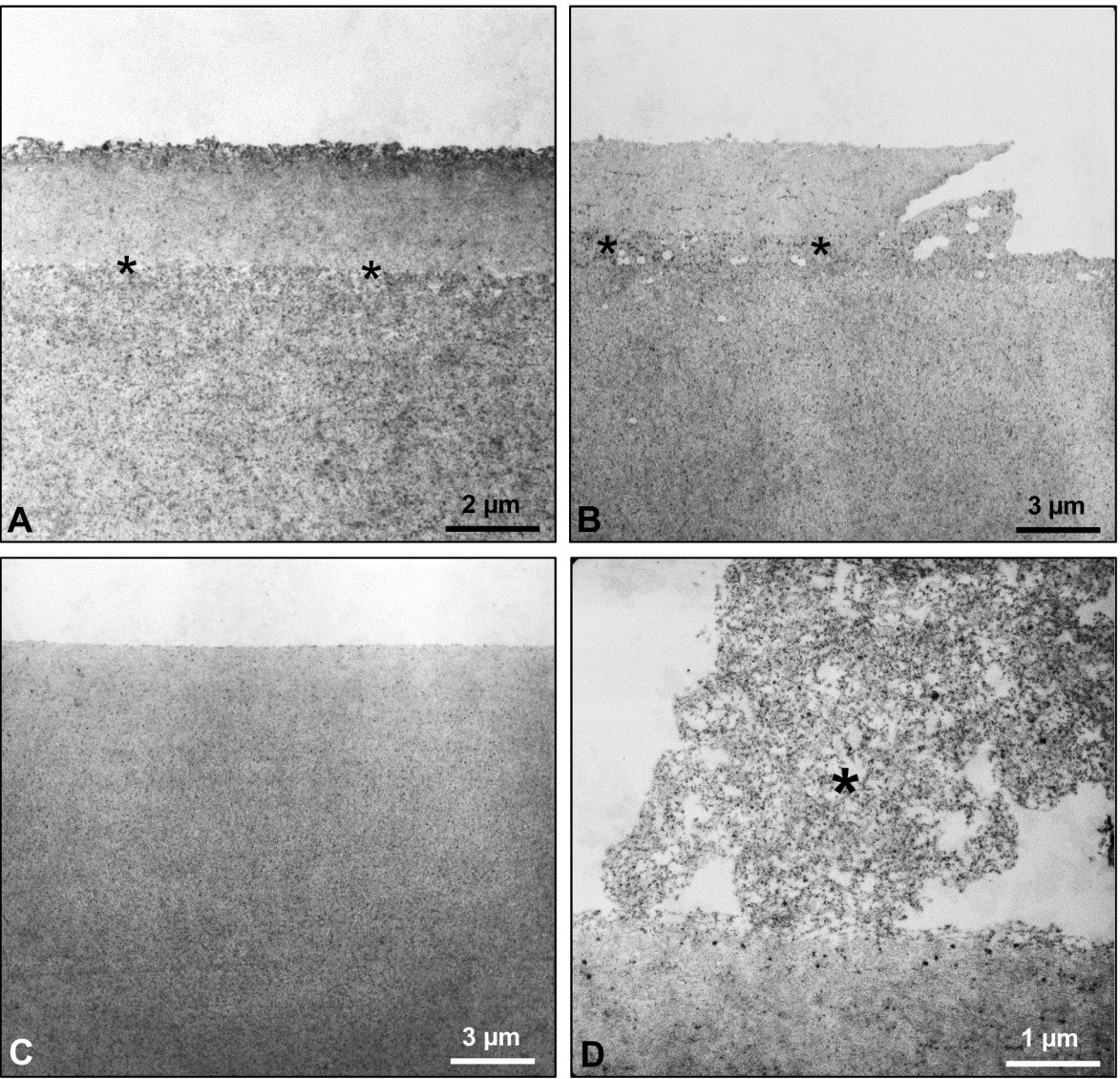

**Fig 6. Transmission electron micrographs of lens capsule specimens showing various alterations of the capsular surface.** (A, B) Layer formation within the superficial capsule with a vacuolar interface (asterisks) between layers. (C) Smooth and compact capsular surface. (D) Precursor PEX material (asterisk) on the capsule surface.

than our findings. This inconsistency can be attributed to the difference in the inclusion criteria for patients with suspected PEX. The criteria in the previous study included the presence of at least two of the following clinical signs: iris atrophy with poor dilated pupils, peripupillary pigment epithelial melanin loss, melanin dispersion after mydriasis, trabecular pigmentation, phacodonesis, and corneal PEX endotheliopathy. Our study included only one sign of anterior lens capsular change, P, C, D, and W, which probably resulted in the inclusion of patients with early-stage disease and led to negative findings of definite PXM. Moreover, the previous study did not mention the tissue preparing protocol. Dark et al. reported that the precapsular film, a ground glass film on anterior lens capsule of patients exhibiting no clinical PXM, is associated with the precursors of PXM; TEM analysis of the precapsular film showed a layer of fine fibrins that were similar in morphology and size to those of PXM [11]. The staining pattern of fibrillin present in

the precapsular film was also similar to that of PXM in PEX [11]. Dark et al. suggested that these precapsular films represent the earliest findings of PEX. In our study, the patients exhibited D, which was similar to that reported in the previous study [11] where no PXM precursors were detected in TEM analysis. Bartholomew reported the pregranular stage, which was also suggested to be the earliest sign of PEX. He found 80 radial, nongranular striae on the mid-third of the anterior lens capsule behind the iris, which morphologically resembled the granular stage of PEX [12]. In our study, anterior lens capsules exhibiting W, which could represent the pregranular stage, showed no association with microscopic findings of PXM. In our study, PXM precursor was detected in one eye exhibiting C, which might be the later stage of early PEX (hole development in intermediate zone) [13]. However, given that only half of the collected capsules were sent for TEM analysis, the analysis of some positive specimens might have been missed.

Interestingly, 59% of the enrolled patients exhibiting pPEX signs showed capsular delamination in LM analysis, which represented TEX. Among them, patients exhibiting C and D showed significant associations with TEX. To the best of our knowledge, our study is the first to report the aforementioned findings in pPEX. Although TEX had previously been considered a rare condition, this was later attributed to underdetection and/or underreporting [14]. Teekhasaenee et al. evaluated a total of 259 patients with TEX who were mostly diagnosed during their seventh decade; aging, intense heat, trauma, inflammation, radiotherapy, and laser iridotomy were identified as the risk factors for disease development, although the disease for most patients was of an idiopathic nature [15, 16]. The incidence of TEX was higher in our study than in a few previous studies. This can be explained on the basis of the meticulous tissue preparation technique used in our study, which increased tissue yield. Considering that capsular delamination usually occurs in circumferential fashion, unevenly but typically beginning at the nasal or temporal quadrant and extending to the inferior and superior quadrants [15], cutting the capsules radially may increase the likelihood of TEX detection. Evidence has shown that TEX is associated with cataract formation, phacodonesis, lens subluxation or dislocation, intraoperative partial capsulorhexis, and glaucoma [15].

In terms of clinical implication, we reported in previous study that the detachment in TEX started along the anterior zonular insertions in association with zonular disruption [16]. Ten percent had spontaneous phacodonesis [15]. Discovering of midperiphery cleft/lacunae and faint central disc present within the photopic pupil preoperatively raises awareness of zonular insufficiency during surgery.

Our study has some limitations. One of these is that PXM detection in LM and TEM analyses depended on the distribution of the material, which was unevenly scattered on the anterior capsules. Differences in the quantity of PXM, disease stage, and topographic relationship between the lens and iris have been identified as the factors associated with PXM detection [1]. Although the capsular delamination in TEX is diffusely located on the anterior lens capsule, a circumferential pattern makes it easier to detect the delamination area. Furthermore, we did not exclude patients with TEX. Preoperative slit-lamp examination detected TEX in only 12% (8 of 66) of the total number of eyes assessed. Most (74%) of the eyes assessed revealed incomplete information regarding the TEX stage because of poorly dilated pupil; excluding TEX diagnosis requires pupillary dilation reaching anterior zonular insertion. In our TEM analysis, the capsules were sectioned into two to three levels for PXM detection. Resectioning into deeper levels was performed only in excised capsule specimens suspected of PXM detection and not in all specimens with negative findings. Prince et al. performed step-sectioning for six steps before marking a specimen as negative [17].

Studies on the mutation of *LOXL1* and immunohistochemistry markers of the anterior lens capsule, such as HNK-1 carbohydrate epitope and fibrillin-1, may facilitate the diagnosis of pPEX [18].

In conclusion, our study found no relationship between pPEX signs and the definite finding of PXM. Interestingly, the detection rate of TEX was more than what we expected, suggesting a strong association between pPEX signs, particularly anterior lens capsules exhibiting C and D, and TEX. However, this might have been a coincidental finding because of the aforementioned limitations. Nevertheless, our study raises awareness regarding TEX diagnosis, particularly in older patients with morphological changes in the anterior lens capsule.

## Acknowledgments

**Meeting presentation:** 46[th] Annual meeting of the Royal College of Ophthalmologist and Ophthalmological Society of Thailand, 2021.

## Author Contributions

**Conceptualization:** Yanin Suwan, Wasu Supakontanasan, Chaiwat Teekhasanee.

**Data curation:** Yanin Suwan, Tuangporn Kulnirandorn, Sattawat Wongchaya, Purit Petpiroon, Apichat Tantraworasin, Chaiwat Teekhasanee.

**Formal analysis:** Yanin Suwan, Apichat Tantraworasin.

**Funding acquisition:** Yanin Suwan.

**Investigation:** Yanin Suwan, Ursula Schlötzer-Schrehardt, Sattawat Wongchaya, Purit Petpiroon, Chaiwat Teekhasanee.

**Methodology:** Yanin Suwan, Tuangporn Kulnirandorn, Ursula Schlötzer-Schrehardt, Purit Petpiroon, Chaiwat Teekhasanee.

**Project administration:** Yanin Suwan.

**Resources:** Purit Petpiroon.

**Supervision:** Yanin Suwan, Wasu Supakontanasan, Chaiwat Teekhasanee.

**Validation:** Yanin Suwan, Chaiwat Teekhasanee.

**Visualization:** Wasu Supakontanasan.

**Writing – original draft:** Yanin Suwan, Tuangporn Kulnirandorn, Wasu Supakontanasan.

**Writing – review & editing:** Yanin Suwan, Ursula Schlötzer-Schrehardt, Purit Petpiroon, Chaiwat Teekhasanee.

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
