## [Decision Letter · Decision Letter 0]

27 Dec 2022

PONE-D-22-32461Light and Electron Microscopic Features of Preclinical Pseudoexfoliation SyndromePLOS ONE

Dear Dr. Suwan,

Thank you for submitting your manuscript to PLOS ONE. After careful consideration, we feel that it has merit but does not fully meet PLOS ONE’s publication criteria as it currently stands. Therefore, we invite you to submit a revised version of the manuscript that addresses the points raised during the review process.

Three learned reviewers offered a number of constructive criticisms designed to improve presentation. These comments can be easily addressed by incorporating appropriate changes in the manuscript.  Please submit your revised manuscript by Feb 10 2023 11:59PM. If you will need more time than this to complete your revisions, please reply to this message or contact the journal office at plosone@plos.org. Please include the following items when submitting your revised manuscript:A rebuttal letter that responds to each point raised by the academic editor and reviewer(s). You should upload this letter as a separate file labeled 'Response to Reviewers'.A marked-up copy of your manuscript that highlights changes made to the original version. You should upload this as a separate file labeled 'Revised Manuscript with Track Changes'.An unmarked version of your revised paper without tracked changes. You should upload this as a separate file labeled 'Manuscript'.If applicable, we recommend that you deposit your laboratory protocols in protocols.io to enhance the reproducibility of your results. Protocols.io assigns your protocol its own identifier (DOI) so that it can be cited independently in the future. For instructions see: https://journals.plos.org/plosone/s/submission-guidelines#loc-laboratory-protocols. Additionally, PLOS ONE offers an option for publishing peer-reviewed Lab Protocol articles, which describe protocols hosted on protocols.io. Read more information on sharing protocols at https://plos.org/protocols?utm_medium=editorial-email&utm_source=authorletters&utm_campaign=protocols.

We look forward to receiving your revised manuscript.

Kind regards,

Sanjoy Bhattacharya

Academic Editor

PLOS ONE

Additional Editor Comments:

All three learned reviewers have requested minor revision that can be done by incorporating appropriate changes in the manuscript

Reviewers' comments:

Reviewer's Responses to Questions

**Comments to the Author**

1. Is the manuscript technically sound, and do the data support the conclusions?

Reviewer #1: Yes

Reviewer #2: Yes

Reviewer #3: Yes

2. Has the statistical analysis been performed appropriately and rigorously? 

Reviewer #1: Yes

Reviewer #2: Yes

Reviewer #3: Yes

3. Have the authors made all data underlying the findings in their manuscript fully available?

Reviewer #1: Yes

Reviewer #2: Yes

Reviewer #3: Yes

4. Is the manuscript presented in an intelligible fashion and written in standard English?

Reviewer #1: Yes

Reviewer #2: Yes

Reviewer #3: Yes

5. Review Comments to the Author

Reviewer #1: Clear and concise paper.

I wish the paper included more information on pPEX findings (P, D, C, W), e.g., some background information and if available, proposed etiology. Why were they specifically chosen? Why were they different than Mardin et al.? If possible, please add information and clarify.

Inclusion/exclusion criteria and methods are well-described.

Minor suggestion and correction:

-Simplify the opening sentence (lines 62-65). Stylistically, it is a bit convoluted and not easy to follow.

-Correct to show* (line 186).

Reviewer #2: Overall, this is an interesting subject and the paper is well-written. The study design is well done, and the introduction provides a thorough background for the study. The pictures provides very helpful additional information and context for the study.

The authors should clarify why only a portion of the samples were sent for TEM analysis (34 samples), and how those specific samples were chosen. The authors should also clarify if any differences were found between the results of the two thicknesses and staining methods (thin samples with toluidine blue, and ultrathin samples with uranyl acetate and lead citrate) used for the samples undergoing TEM analysis.

It would be helpful if the authors could elaborate on the clinical significance of their findings, that pPEX is not associated with PXM and is significantly associated with TEX.

Reviewer #3: The study evaluated the ultrastructural alterations of the anterior capsules in patients with signs of preclinical pseudoexfoliation syndrome using histological analyses with light microscopy and transmission electron microscopy in 34 pPEX patients and 62 controls undergoing cataract surgeries in Thailand. The manuscript is well-written with clear logics. The presented data was self-explaining and easy to follow. The authors explained the possible reasons leading to the different outcomes of this study compared to several published reports. The findings here could contribute to the diverse clinical studies in identifying the subclinical features related to PEX diagnosis. The study could be strengthened in a few ways.

1. The introduction justified the importance to diagnose PEX in its early stage. It will be more helpful to add more information about the prevalence of PEX worldwide and the percentage of PEX patients developing elevated IOP and glaucoma eventually. This could help readers understand the different stages of the disease - PEX.

2. TEX diagnosis has been a major finding in this study. However, its diagnosis criteria needs to be detailed in the methods section. Is it necessary to remove any patients with clear clinical signs of TEX in the first place of study enrollment?

3. It has been more than two years after the initial study. Is it possible to check whether any of the study patients has developed clinical symptoms of PEX, especially the one case with TEM alterations?

6. PLOS authors have the option to publish the peer review history of their article (what does this mean?). If published, this will include your full peer review and any attached files.

Reviewer #1: No

Reviewer #2: No

Reviewer #3: No

---

## [Author Response · Author response to Decision Letter 0]

28 Jan 2023

Dear Prof. Sanjoy Bhattacharya,

The authors would like to thank you and the reviewers for the support and thoughtful critique of our manuscript. Please find attached a revised version of the manuscript (PONE-D-22-32461) entitled “Light and Electron Microscopic Features of Preclinical Pseudoexfoliation Syndrome”. Our responses to the reviewers’ comments are shown in red and enclosed below. We hope that these changes strengthen the manuscript so that it may be accepted for eventual publication in PLOS ONE. We look forward to hearing from you and the reviewers, and please feel free to contact us with any comments or questions.

Best regards,

Yanin Suwan, MD

Department of Ophthalmology

Ramathibodi Hospital, Mahidol University

Rama VI, Bangkok

Thailand, 10400.

Email: yanin.suwan@gmail.com

Additional Editor Comments:

All three learned reviewers have requested minor revision that can be done by incorporating appropriate changes in the manuscript

Reviewers' comments:

Reviewer's Responses to Questions

Comments to the Author

1. Is the manuscript technically sound, and do the data support the conclusions?

Reviewer #1: Yes

Reviewer #2: Yes

Reviewer #3: Yes

2. Has the statistical analysis been performed appropriately and rigorously? 

Reviewer #1: Yes

Reviewer #2: Yes

Reviewer #3: Yes

3. Have the authors made all data underlying the findings in their manuscript fully available?

Reviewer #1: Yes

Reviewer #2: Yes

Reviewer #3: Yes

4. Is the manuscript presented in an intelligible fashion and written in standard English?

Reviewer #1: Yes

Reviewer #2: Yes

Reviewer #3: Yes

5. Review Comments to the Author

Reviewer #1: Clear and concise paper.

I wish the paper included more information on pPEX findings (P, D, C, W), e.g., some background information and if available, proposed etiology. Why were they specifically chosen? Why were they different than Mardin et al.? If possible, please add information and clarify. 

Thank you for your comments. I added the proposed etiology of each sign. The differences between our study and Mardin et al. had been described in discussion part. 

Inclusion/exclusion criteria and methods are well-described.

Minor suggestion and correction:

-Simplify the opening sentence (lines 62-65). Stylistically, it is a bit convoluted and not easy to follow.

Thank you for your comment. Please see the correction in L62-64. 

-Correct to show* (line 186).

Thank you for your comment. I made the correction as per your suggestion.

Reviewer #2: Overall, this is an interesting subject and the paper is well-written. The study design is well done, and the introduction provides a thorough background for the study. The pictures provides very helpful additional information and context for the study.

The authors should clarify why only a portion of the samples were sent for TEM analysis (34 samples), and how those specific samples were chosen. 

Some of the specimens had been lost and fallen dry during transportation to the lab. 

The authors should also clarify if any differences were found between the results of the two thicknesses and staining methods (thin samples with toluidine blue, and ultrathin samples with uranyl acetate and lead citrate) used for the samples undergoing TEM analysis.

The two methods have strong correlation. If ultrathin sectioning missed the lesion that had been observed by thin sectioning, the procedure was repeated.

It would be helpful if the authors could elaborate on the clinical significance of their findings, that pPEX is not associated with PXM and is significantly associated with TEX.

Thank you for your comment. I added the clinical significance of our findings in L279-283.

Reviewer #3: The study evaluated the ultrastructural alterations of the anterior capsules in patients with signs of preclinical pseudoexfoliation syndrome using histological analyses with light microscopy and transmission electron microscopy in 34 pPEX patients and 62 controls undergoing cataract surgeries in Thailand. The manuscript is well-written with clear logics. The presented data was self-explaining and easy to follow. The authors explained the possible reasons leading to the different outcomes of this study compared to several published reports. The findings here could contribute to the diverse clinical studies in identifying the subclinical features related to PEX diagnosis. The study could be strengthened in a few ways.

1. The introduction justified the importance to diagnose PEX in its early stage. It will be more helpful to add more information about the prevalence of PEX worldwide and the percentage of PEX patients developing elevated IOP and glaucoma eventually. This could help readers understand the different stages of the disease - PEX.

Thank you for your comments. Please see the correction in L71-76. I prefer not to mention the reported prevalence of PEX since it varies significantly due to lack of uniformly conducted epidemiologic studies.

2. TEX diagnosis has been a major finding in this study. However, its diagnosis criteria needs to be detailed in the methods section. Is it necessary to remove any patients with clear clinical signs of TEX in the first place of study enrollment?

Thank you for your comment. I totally agree with your concern. Eyes with preoperative finding of TEX should be excluded. However, the purpose of this study is to explore the preclinical sign of PEX. We didn’t expect the association of these signs with TEX in the first place. In addition, early diagnosis of TEX needs fully dilated eye examination to see anterior zonular disinsertion. Most (74%) of the eyes assessed revealed incomplete information regarding the TEX stage because of poorly dilated pupil.

3. It has been more than two years after the initial study. Is it possible to check whether any of the study patients has developed clinical symptoms of PEX, especially the one case with TEM alterations?

Thank you for the comment. I checked each patient on the last follow. Since patients were seen postoperatively by several doctors, PEX material were not carefully evaluated and specifically mentioned. 

6. PLOS authors have the option to publish the peer review history of their article (what does this mean?). If published, this will include your full peer review and any attached files.

Do you want your identity to be public for this peer review? For information about this choice, including consent withdrawal, please see our Privacy Policy.

Reviewer #1: No

Reviewer #2: No

Reviewer #3: No

---

## [Decision Letter · Decision Letter 1]

23 Feb 2023

Light and Electron Microscopic Features of Preclinical Pseudoexfoliation Syndrome

PONE-D-22-32461R1

Dear Dr. Suwan,

We’re pleased to inform you that your manuscript has been judged scientifically suitable for publication and will be formally accepted for publication once it meets all outstanding technical requirements.

Kind regards,

Sanjoy Bhattacharya

Academic Editor

PLOS ONE

Additional Editor Comments (optional):

Reviewers' comments:

Reviewer's Responses to Questions

**Comments to the Author**

1. If the authors have adequately addressed your comments raised in a previous round of review and you feel that this manuscript is now acceptable for publication, you may indicate that here to bypass the “Comments to the Author” section, enter your conflict of interest statement in the “Confidential to Editor” section, and submit your "Accept" recommendation.

Reviewer #1: All comments have been addressed

Reviewer #2: All comments have been addressed

Reviewer #3: All comments have been addressed

2. Is the manuscript technically sound, and do the data support the conclusions?

Reviewer #1: Yes

Reviewer #2: Yes

Reviewer #3: Yes

3. Has the statistical analysis been performed appropriately and rigorously? 

Reviewer #1: Yes

Reviewer #2: Yes

Reviewer #3: Yes

4. Have the authors made all data underlying the findings in their manuscript fully available?

Reviewer #1: Yes

Reviewer #2: Yes

Reviewer #3: Yes

5. Is the manuscript presented in an intelligible fashion and written in standard English?

Reviewer #1: Yes

Reviewer #2: Yes

Reviewer #3: Yes

6. Review Comments to the Author

Reviewer #1: All comments were addressed in the text. The paper is well-written, and the figures clearly demonstrate the pPEX features.

Reviewer #2: The reviewer comments have adequately been addressed in this revised submission for this manuscript.

Reviewer #3: The authors have addressed all the comments from last round. The quality of the manuscript has been improved. No additional comments.

7. PLOS authors have the option to publish the peer review history of their article (what does this mean?). If published, this will include your full peer review and any attached files.

Reviewer #1: No

Reviewer #2: No

Reviewer #3: No

---

## [Editor Report · Acceptance letter]

28 Feb 2023

PONE-D-22-32461R1 

Light and Electron Microscopic Features of Preclinical Pseudoexfoliation Syndrome 

Dear Dr. Suwan:

I'm pleased to inform you that your manuscript has been deemed suitable for publication in PLOS ONE. Congratulations! Your manuscript is now with our production department. 

Kind regards, 

on behalf of

Dr. Sanjoy Bhattacharya 

Academic Editor

PLOS ONE